# Local infiltration analgesia with bupivacaine and adrenaline does not reduce perioperative blood loss in total hip arthroplasty

**Marcin Ceynowa**[1]*, **Tomasz Sikora**[2], **Marek Rocławski**[1], **Mariusz Treder**[1], **Krzysztof Kolarz**[1], **Rafał Pankowski**[1], **Tomasz Mazurek**[1]

**1** Department of Orthopedic Surgery, Medical University of Gdańsk, Gdańsk, Poland, **2** Department of Orthopedic Surgery, Staszów County Hospital, Staszów, Poland

* mceynowa@gumed.edu.pl

## Abstract

This study evaluates the effect of local infiltration analgesia with bupivacaine and adrenaline on perioperative blood loss in total hip arthroplasty. Patients who had primary total hip arthroplasty were retrospectively assigned to two groups. One group had 100 ml of bupivacaine/adrenaline solution injected into periarticular soft tissues at the end of the procedure. There were 55 patients in the infiltrated hip group and 44 patients in the not infiltrated group. Patients' hemoglobin level (Hb), hematocrit (HTC), red blood count (RBC), platelet count (PLT) and International Normalized Ratio (INR) as well as the need for blood transfusions were compared statistically between groups preoperatively and postoperatively. There were no significant differences between Hb, HTC or RBC levels as well as the rate and amount of blood transfusions on the 1st, 4th postoperative days or at patients' discharge between infiltrated and not infiltrated groups. This study does not support the hypothesis that the use of local infiltration analgesia with adrenaline may reduce perioperative blood loss in total hip arthroplasty.

## Introduction

Hip arthroplasty is one of the most common orthopedic surgeries performed worldwide [1–3]. It is an extensive procedure with a range of associated problems, including pain and significant blood loss, among other [1, 3, 4]. Many improvements have been made in the operative technique [2, 5, 6] as well as in the perioperative management of the patient to minimize operative trauma and improve rehabilitation process [1, 4, 7].

Local infiltration analgesia (LIA) is a well-recognized method for controlling pain in the early postoperative period. In orthopedic surgery, it is most commonly used for total hip and knee arthroplasty [4, 8, 9]. A range of different solutions of regional anaglesics are used diluted in normal saline [4], with or without addition of adrenaline. Adrenaline in this solution has proven to have an additional benefit of decreasing perioperative blood loss and transfusion rates in total knee arthroplasty [10], especially when used together with tranexamic acid [11]. In total hip arthroplasty, local infiltration analgesia with adrenaline has proven to be beneficial

**Data Availability Statement:** All relevant data are within the paper and its Supporting Information files.

**Funding:** The author(s) received no specific funding for this work.

**Competing interests:** The authors have declared that no competing interests exist.

in postoperative pain control [7, 12], but it failed to show a consistent effect on bleeding [3, 13, 14] using different solutions and application protocols as in this study.

Our hypothesis was that local infiltration analgesia with bupivacaine and adrenaline in total hip arthroplasty would significantly decrease the perioperative blood loss in patients who were infiltrated with the solution by at least 350 ml (what is approximately equal to 1 unit of transfused blood) [10], what would in turn reduce the required amount of blood transfusion. This study attempts to answer the question whether LIA with adrenaline can be administered for the purpose of limiting perioperative blood loss, what would provide additional benefit when compared to LIA without adrenaline or to local nerve blocks.

## Material and methods

All patients who underwent standard total hip arthroplasty between 1. October 2009 and 30. May 2010 in the Department of Orthopedic Surgery, Medical University of Gdansk, Poland were retrospectively analysed in the study. Patients provided an informed written consent to allow their medical data to be analyzed in this study. This study was approved by the local Bioethics Committee (issued 25.11.2009, NKEBN/332/2009). Only patients with primary osteoarthritis were included. Total hip arthroplasty is a standard procedure for treatment of this disease at our institution. No modifications in patient treatment were undertaken for the purpose of this study, and the patient's standard medical data was analysed retrospectively. All patients were prepared for surgery by a general practitioner and consulted by an anaesthesiologist to ensure that their potential comorbidities, including hypertension, diabetes mellitus, coronary heart disease or other less common diseases, were well controlled and the general risk of surgery regarding overall health status was minimized. Exclusion criteria were as follows: previous fractures of the femur or acetabulum, a history of septic arthritis, previous hip surgery, osteonecrosis, coagulopathy, or general contraindications for infiltration with local anesthetic and adrenaline.

Patients were operated by senior residents (MC, TS, MR), under the supervision of two senior orthopedic surgeons (KK, MT). In Poland, patients can choose any hospital they want to be treated in and once they choose they become assigned to the hospital's orthopedic centre, but not to a particular surgeon in this hospital. Therefore, the patients were assigned at random to one of the senior surgeons for supervision and one of the senior residents for surgery. Once the patient has given an informed consent regarding surgery as well as accepted the operating team, the operation was performed. One of the senior surgeons (MT) required a routine use of infiltration analgesia during surgeries that he supervised for the purpose of early postoperative pain control, the other (KK) did not used it as a part of standard patients' care. This created two groups of patients, where one group received infiltration analgesia and the other did not. This approach, based on supervising surgeon preference regarding infiltration analgesia was a standard practice at our institution until 2013, when local nerve blocks were introduced and wound infiltration with local anesthetics was no longer used for early postoperative pain control. During patients' hospital stay, their clinical data was collected including gender, age, weight, accompanying chronic ailments, the size of prosthesis components as well as the drainage output and blood parameters measured pre and postoperatively.

Patients were operated on using standard spinal anesthesia [10]. The standard lateral Hardinge approach was used for the hip arthroplasty [15]. The local anesthetic mixture was prepared with 100 ml saline solution with 50 mg of bupivacaine and 1 mg adrenaline. This solution was prepared according to a standard developed in our institution, and is similar to the solutions used in other studies [3]. The infiltration analgesia was used after completion of the arthroplasty procedure, but before wound closure. The solution was injected into the

tissues around the rim of the acetabulum including the capsule, the exposed incised gluteus medius and vastus lateralis muscles, as well as into the subcutaneous tissue. A total of 100 ml of solution was used. Two drains were inserted: one into the joint, and the other under the fascia lata after gluteus medius closure. The operative time was recorded within 5 minute intervals.

Drainage was continued for 48 hours after surgery. The drainage output was collected with Redon drains, the volume was recorded after 24 and 48 hours separately. Hemoglobin levels (Hb), hematocrit (HTC) and red blood cell count (RBC) was measured after 24 hours (Day 1) and on the 4th postoperative day (Day 4). Blood transfusion was indicated when Hb levels were below 10 g/100 ml on Day 1, or 9 g/100 ml on the Day 4. Patients were discharged home on Day 4. In patients who required blood transfusions on Day 4 as well as the patients who had Hb levels between 9 and 10 g/100 ml, the discharge was delayed for 48 hours when the blood parameters were measured once again and then they were discharged. Therefore, the Hb, HTC and RBC at discharge are a pooled data of their final levels at the end of hospital stay, where the Day 4 levels of patients who had their hospital stay prolonged were substituted for their final levels at discharge.

Patients were taught to get out of bed and walk with crutches on the first or second postoperative day, depending on their functional status, in physiotherapist's assistance. The postoperative pain and functional status was not recorded and assessed in detail.

Patients' gender, age, weight, body mass index (BMI), acetabulum and stem size, operative time, Hb, HTC, RBC, as well platelet count (PLT) and International Normalized Ratio (INR) to screen for bleeding disorders were compared statistically between groups. The blood volume was estimated from the preoperative Hb levels, body weight and height using the formula from Meunier et al. [10, 16] and compared as well. There were 55 patients in the infiltrated hip group (M:F = 22:33) and 44 patients in the not infiltrated group (M:F = 24:20). The data on implant size was available in 40 patients in the infiltrated group and in 37 patients in the not infiltrated group. Patients' characteristics and comparison between groups are given in Table 1. The comparison of Hb, HTC, RBC and INR preoperatively is given in Table 2. All patients' data is supplied in the S1 File.

The sample size calculation was performed considering a 350 ml (SD = 300 ml) difference in blood loss to be significant [10] at an average total blood loss of 1200 ml. Alpha and beta levels were set at 5% and 20%, respectively. The required total sample size was 24 (12 per group). In other similar studies, a maximum total number of enrolled patients required was 70 with 100 tested [10], 12 with 49 tested [10], 20 with 121 tested [17]. Current study had a total of 99 patients (55 v/s 44 in both groups), therefore it should be considered adequately powered.

The Kolmogorov-Smirnov test was used to assess normality of distribution. The two-tail T-student test was used for comparison between results was used when normal distribution was assumed, and the Mann Whitney U test was used for comparison of results when normal distribution was not assumed. The chi-square test was used for comparison between groups when categorical variables were analysed. The Statistica PL v. 13.3 software was used for calculations.

## Results

There no significant differences in the female to male ratio in both groups (2x2 chi-square test p = 0.15), as well as with regard to age, weight, BMI, estimated blood volume, Hb, HTC, RBC, PLT or INR between corresponding groups (Tables 1 and 2). The acetabular size was slightly, but significantly larger in the not infiltrated group, with no differences in stem size (Table 1).

**Table 1. Patient characteristics.**

| | Infiltrated group | Not infiltrated group | Statistics |
|---|---|---|---|
| | (n = 55) | (n = 44) | |
| Age (years) | 68.07 | 68.05 | p = 0.98 |
| | SD = 9.3 | SD = 8.9 | |
| | R: 46–86 | R: 47–87 | |
| Weight (kilograms) | 78.4 | 77.3 | p = 0.69 |
| | SD = 15.1 | SD = 13.7 | |
| | R: 48–130 | R: 55–109 | |
| Body mass index | 26.46 | 25.94 | p = 0.23 |
| | SD = 3.32 | SD = 2.54 | |
| | R: 20–38 | R: 21–32 | |
| Acetabulum size[a] (mm) | 54.95 | 55.38 | p < 0.001 |
| | SD = 4.19 | SD = 4.78 | |
| | R: 46–62 | R: 44–65 | |
| Stem size[a] | 12.19 | 11.75 | p = 0.63 |
| | SD = 2.54 | SD = 2.68 | |
| | R: 7.5–18 | R: 7.5–17.5 | |
| Estimated blood volume (ml) | 2914.63 | 2928.02 | p = 0.76 |
| | SD = 592.45 | SD = 445.6 | |
| | R: 1770.9–4483.46 | R: 2002.46–4112.78 | |

[a] The sample sizes for the acetabulum and the stem were n = 40 patients in the infiltrated group and n = 37 patients in the not infiltrated group.

**Table 2. The comparison of preoperative blood and INR results.**

| | Infiltrated group | Not infiltrated group | Statistics |
|---|---|---|---|
| | (n = 55) | (n = 44) | |
| Hb (g/dl) | 13.9 | 13.94 | p = 0.33 |
| | SD = 1.35 | SD = 1.2 | |
| | R: 11.5–16.5 | R: 11.5–16.4 | |
| HTC (%) | 41.4 | 41 | p = 0.91 |
| | SD = 3.5 | SD = 2.9 | |
| | R: 35.1–49.1 | R: 36–48 | |
| RBC (T/l) | 4.68 | 4.7 | p = 0.46 |
| | SD = 0.38 | SD = 0.39 | |
| | R: 4.06–5.64 | R: 4.02–5.46 | |
| PLT (G/l) | 253 | 244 | p = 0.52 |
| | SD = 77 | SD = 69 | |
| | R: 153–456 | R: 140–425 | |
| INR | 1.03 | 1.03 | p = 0.73 |
| | SD = 0.09 | SD = 0.1 | |
| | R: 0.85–1.25 | R: 0.82–1.32 | |

Hb, hemoglobin level; HTC, hematocrit; RBC, red blood count; PLT, platelet count; INR, International Normalized Ratio.

**Table 3. The comparison of postoperative hemoglobin levels (Hb), hematocrit (HTC) and red blood count (RBC).**

| Postoperative day | | Infiltrated group | Not infiltrated group | Statistics |
|---|---|---|---|---|
| | | (*n* = 55) | (*n* = 44) | |
| Day 1 | Hb (g/dl) | 11.24 | 11.55 | p = 0.92 |
| | | SD = 1.2 | SD = 1.34 | |
| | | R: 9–14.1 | R: 7.6–14.4 | |
| | HTC (%) | 33.3 | 34 | p = 0.95 |
| | | SD = 3.43 | SD = 3.8 | |
| | | R: 26.3–42.4 | R: 23–41 | |
| | RBC (T/l) | 3.81 | 3.89 | p = 0.80 |
| | | SD = 0.4 | SD = 0.48 | |
| | | R: 3.04–5.07 | R: 2.51–4.89 | |
| Day 4 | Hb (g/dl) | 10.64 | 10.73 | p = 0.83 |
| | | SD = 1.36 | SD = 1.22 | |
| | | R: 7.2–13.9 | R: 8.3–13.6 | |
| | HTC (%) | 31.6 | 32 | p = 0.89 |
| | | SD = 3.86 | SD = 3.4 | |
| | | R: 20.4–41.8 | R: 24–40 | |
| | RBC (T/l) | 3.58 | 3.66 | p = 0.96 |
| | | SD = 0.48 | SD = 0.45 | |
| | | R: 2.36–5.14 | R: 2.65–4.89 | |
| Discharge | Hb (g/dl) | 10.91 | 10.97 | p = 0.71 |
| | | SD = 1.08 | SD = 1.12 | |
| | | R: 9.1–13.9 | R: 9.3–13.6 | |
| | HTC (%) | 32.5 | 33 | p = 0.97 |
| | | SD = 2.9 | SD = 3.1 | |
| | | R: 27.4–41.8 | R: 28–40 | |
| | RBC (T/l) | 3.68 | 3.74 | p = 0.99 |
| | | SD = 0.38 | SD = 0.41 | |
| | | R: 3.02–5.14 | R: 2.97–4.89 | |

Hb, hemoglobin level; HTC, hematocrit; RBC, red blood count.

No significant differences were found regarding operative time in both groups (p = 0.47). In the infiltrated group it was 90.81 minutes (SD– 15.06; R: 55–125), in the not infiltrated group it was 90.56 minutes (SD = 14.38, R: 60–120).

There were no significant differences between Hb, HTC or RBC levels on Day 1, Day 4 or at patients' discharge between infiltrated and not infiltrated groups (Table 3).

The drain output (Table 4) was significantly greater (T-student test) in the not infiltrated group on the 2$^{nd}$ postoperative day, as well as the total output (the sum of the 1$^{st}$ and 2$^{nd}$ post-operative day). The differences were on average 31 ml on the 1$^{st}$ day, 34 ml on the 2$^{nd}$ and 65 ml in total.

Since the acetabular size was significantly greater in the not infiltrated group, the results were corrected for the acetabular size according to the following simple formula developed by the authors:

Ratio = drainage output volume (ml)/acetabular size (mm)

After correction, the drainage output showed no significant differences between groups for the 1$^{st}$ postoperative day, 2$^{nd}$ postoperative day or total output (Table 5). A strong positive

**Table 4. Drainage output volume in milliliters.**

| Postoperative day | Infiltrated group | Not infiltrated group | Statistics |
|---|---|---|---|
| | (*n* = 55) | (*n* = 44) | |
| Day 1 | 365 | 396 | p = 0.24 |
| | SD = 127 | SD = 134 | |
| | R: 80–600 | R: 70–720 | |
| Day 2 | 156 | 190 | p = 0.025 |
| | SD = 70 | SD = 78 | |
| | R: 30–360 | R: 80–450 | |
| Total | 521 | 586 | p = 0.04 |
| | SD = 150 | SD = 156 | |
| | R: 190–810 | R: 340–940 | |

correlation (Pearson correlation) was found between acetabular size and bleeding (R: 0.94, p<0.001), but for the stem size there was a weak negative correlation (R: -0.3, p = 0.006).

The infiltrated hip group required on average 1.53 blood units (SD = 1.24, R: 0–5) to be transfused and the not infiltrated group 1.61 (SD = 1.11, R: 0–4). The differences were statistically not significant (T-test p = 0.71). In the infiltrated group, 18 of 55 patients did not require transfusion, an in the not infiltrated group, 12 of 44 patients (chi-square test p = 0.39).

The blood loss was estimated according to Meunier et al. [10, 16]. The blood loss was compared in the 1st postoperative day before any blood products were transfused. Moreover, taking into account the assumption that one unit of transfused blood equates to approximately 350ml of blood, total blood loss was estimated with the previous mentioned formula using Hb levels at discharge. The estimated volume of transfused blood was added by multiplying the number of blood units by 350 ml. The 1st day and total blood loss was compared between groups, and no significant differences were found (Table 6).

The total estimated blood loss was corrected for acetabular size in a similar fashion as the drain output shown in Table 5, using the formula:

Ratio = total blood loss (ml)/acetabular size (mm).

In the infiltrated group (n = 40) the ratio was 21.41 (SD = 7.99; R: 10.1–45.26), in the not infiltrated group (n = 37) the ratio was 21.37 (SD = 6.04; R: 11.77–34.6). The results were statistically not significant (p = 0.47).

A separate analysis in two age groups: 60–69 and 70–89 years of age was performed additionally (S2 File). The only significant difference found was regarding acetabular size in the

**Table 5. Drainage output volume in milliliters corrected for acetabular size in millimeters (ratio: Drainage output volume/acetabular size).**

| Postoperative day | Infiltrated group | Not infiltrated group | Statistics |
|---|---|---|---|
| | (*n* = 55) | (*n* = 44) | |
| Day 1 | 6.76 | 7.21 | p = 0.86 |
| | SD = 2.23 | SD = 2.04 | |
| | R: 1.66–10.64 | R: 1.89–12 | |
| Day 2 | 2.86 | 3.46 | p = 0.31 |
| | SD = 1.26 | SD = 1.43 | |
| | R: 0.52–6 | R: 1.53–8.33 | |
| Total | 9.62 | 10.66 | p = 0.19 |
| | SD = 2.58 | SD = 2.45 | |
| | R: 4.1–15.38 | R: 6.55–16.66 | |

**Table 6.  Comparison of estimated blood loss in milliliters between groups.**

|          | Infiltrated group | Not infiltrated group | Statistics |
|----------|-------------------|-----------------------|------------|
|          | (*n* = 55)        | (*n* = 44)            |            |
| Day 1    | 549.41            | 495.32                | p = 0.37   |
|          | SD = 268.28       | SD = 284.97           |            |
|          | R: 60.94–1232.29  | R: 67.22–1138.24      |            |
| Total    | 1158.22           | 1191.82               | p = 0.84   |
|          | SD = 425.78       | SD = 367.1            |            |
|          | R: 525.1–2410.23  | R: 544.1–2006.17      |            |

70–79 years old group, but the correction performed in a similar manner for drainage output and total blood loss did not show any statistically significant differences. No significant differences between the infiltrated and not infiltrated groups were found in any other evaluated parameter.

No incident of deep vein thrombosis, infection, acute cardiac arrhythmia, allergic reactions or hip dislocation occurred in either groups during hospital stay.

## Discussion

This study failed to prove any significant benefit of periarticular injection of a solution of a local anesthetic and adrenaline in total hip arthroplasty. Both the infiltrated and the not infiltrated groups had a similar loss of Hb, HTC and RBC levels, a similar number of patients that required blood transfusions, as well as there was the same average number of blood unit per patient. The estimated blood loss did not differ significantly between groups. There was a slightly greater drain output in the not-infiltrated group, but when the results were corrected for acetabular size, no significant differences were found. A similar correction performed for total estimated blood loss did not show any significant differences either. A separate evaluation in age groups showed similar results as in the comparison of the whole groups.

Interestingly, there was a significant correlation found between implant size and drain output. This suggests that acetabular size, but not necessarily the femoral stem size, may be a significant predictor of perioperative bleeding. Such findings were not reported in literature before. However, since this factor proved to be significant, it should be included in similar studies in the future, as the difference in prosthesis size between groups may bias the outcome. The findings of this study are with concordance with a randomized controlled study on the effect of local infiltration analgesia on acute pain and bleeding after primary total hip arthroplasty by Villatte et al. [3], who found no differences in Hb levels or the need of blood transfusions in either groups.

A study by Yewlett et al. [14] showed that washing the surgical field with a saline solution with epinephrine does not reduce intraoperative blood loss, drainage levels or drop in haemoglobin. However, they found that the patients who had the surgical field washed with epinephrine solution required significantly less blood transfusions, what is different from our study.

This study is in contrast with the finding that a solution of bupivacaine with epinephrine similar to the one used in the current study reduce blood loss in total knee arthroplasty [18]. Generally, solutions containing adrenaline were found to reduce perioperative blood loss in total knee arthroplasty [10, 19, 20], with some exceptions [11, 17].

Several studies suggest that solutions which include tranexamic acid with adrenaline have a significant effect in controlling blood loss in total hip arthroplasty [1]. Our study, which did not include tranexamic acid, failed to prove any benefit. Those differences in findings are in

concordance with the findings of Durgut et al. [11], who found that adrenaline alone does not decrease bleeding, but the tranexamic acid does.

This study does not assess pain intensity or functional status, what may be considered a limitation. Those parameters are sometimes evaluated in similar studies [3], but are not required to draw conclusions about the bleeding status [10, 11, 21]. The observation period included only the duration of hospital stay [10, 11, 21]. Longer periods of observation are required to evaluate the analgesic effect of the local infiltration analgesia [3, 9]. Another limitation is that the patients were not operated by the same surgeon, but by the same senior residents in both groups with a similar experience in performing hip arthroplasty. Operative time in both groups was similar. Therefore the surgeons' experience and technique therefore would likely not affect the results of the study. The perioperative care was the same for both groups, as they were operated on in the same period of time in a single orthopedic clinic [10].

In conclusion, this study does not support the hypothesis that the use of local infiltration analgesia with adrenaline may reduce perioperative blood loss in total hip arthroplasty, therefore its use for the purpose of limiting blood loss or reduction of transfusion rates is unjustified. Moreover, this study found that prosthesis size, especially the acetabulum size, may be a significant predictor of blood loss. This factor should be taken into account in similar studies in the future.

## Supporting information

**S1 File. Data file.**
(XLSX)

**S2 File. Supplementary tables file.**
(DOCX)

## Author Contributions

**Conceptualization:** Marcin Ceynowa, Tomasz Sikora, Tomasz Mazurek.

**Data curation:** Marcin Ceynowa, Tomasz Sikora, Marek Rocławski, Mariusz Treder, Krzysztof Kolarz.

**Formal analysis:** Marcin Ceynowa, Tomasz Sikora, Rafał Pankowski.

**Investigation:** Marcin Ceynowa, Tomasz Sikora, Marek Rocławski, Mariusz Treder, Krzysztof Kolarz.

**Methodology:** Marcin Ceynowa, Tomasz Sikora, Marek Rocławski, Rafał Pankowski.

**Project administration:** Marcin Ceynowa.

**Resources:** Marcin Ceynowa, Krzysztof Kolarz, Tomasz Mazurek.

**Supervision:** Marcin Ceynowa, Mariusz Treder, Krzysztof Kolarz, Tomasz Mazurek.

**Validation:** Marcin Ceynowa, Tomasz Sikora, Marek Rocławski, Mariusz Treder, Krzysztof Kolarz, Rafał Pankowski.

**Visualization:** Tomasz Sikora, Marek Rocławski, Mariusz Treder, Rafał Pankowski.

**Writing – original draft:** Marcin Ceynowa, Tomasz Sikora, Marek Rocławski, Mariusz Treder, Krzysztof Kolarz, Tomasz Mazurek.

**Writing – review & editing:** Marcin Ceynowa, Tomasz Sikora, Marek Rocławski, Mariusz Treder, Rafał Pankowski, Tomasz Mazurek.

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
