## [Decision Letter · Decision Letter 0]

3 Jun 2021

PONE-D-21-08207

Local infiltration analgesia with bupivacaine and adrenaline does not reduce perioperative blood loss in total hip arthroplasty.

PLOS ONE

Dear Dr. Ceynowa,

Thank you for submitting your manuscript to PLOS ONE. After careful consideration, we feel that it has merit but does not fully meet PLOS ONE’s publication criteria as it currently stands. Therefore, we invite you to submit a revised version of the manuscript that addresses the points raised during the review process.

Please address the issues and revise accordingly.

We look forward to receiving your revised manuscript.

Kind regards,

Academic Editor

PLOS ONE

Journal Requirements:

Reviewers' comments:

Reviewer's Responses to Questions

**Comments to the Author**

1. Is the manuscript technically sound, and do the data support the conclusions?

Reviewer #1: Yes

Reviewer #2: No

2. Has the statistical analysis been performed appropriately and rigorously? 

Reviewer #1: I Don't Know

Reviewer #2: No

3. Have the authors made all data underlying the findings in their manuscript fully available?

Reviewer #1: Yes

Reviewer #2: Yes

4. Is the manuscript presented in an intelligible fashion and written in standard English?

Reviewer #1: Yes

Reviewer #2: Yes

5. Review Comments to the Author

Reviewer #1: General comments

As long as wound infiltration with local anesthetics is no longer necessary for early postoperative pain control after development of local nerve blocks (lines 70-72),\\n what is the importance of the current research?

Abstract

. Line 23: meaning of the abbreviations Hb, HTC or RBC \\nwere not previously reported

. Lines 24-25: “amount of blood transfusions on the 1st, 4th or at patients’ discharge between infiltrated and not infiltrated groups”. \\nshould be corrected to” the 1st or 4th post-operative days”

Introduction

. Introduction must highlight the importance of the current research. The authors in lines 42-44: In total hip arthroplasty, local infiltration analgesia with adrenaline has proven to be \\nbeneficial in postoperative pain control [7,12], but it failed to show a consistent effect on bleeding [3,13,14]. \\nSo, what is the clinical significance of the current research?

. Material and Methods

The 2 groups are not operated by the same surgeon and this is one of the limitations of the study, it is not a randomization and not a point of strength

Line 99: Meaning of abbreviation “BMI” was not previously mentioned.

Results

. Line 137: “There no significant differences in the female to male ratio in both groups (2x2 chi-square test “\\n

. Authors did not report use of Chi – square test in Statistical methods (lines 130-133)

. Line 169: “Ratio = drainage output volume (ml)/acetabular size (mm)” Was this equation developed by the authors? \\n

Discussion

Is too long and should be shortened

. Line 226: “The study is not a randomized controlled study, but some criteria of randomization are met”. This sentence is not true and must be deleted. \\n

Tables

Considering each table as a separate entity, meaning of all abbreviations used in each table must be reported in a footnote after each table.

Reviewer #2: The authors evaluated whether local infiltration analgesia with bupivacaine and adrenaline reduced perioperative blood loss in THA. They compared a group with the analgesia and the other without it in a retrospective manner and found that there were no significant differences between the two groups. This is an interesting study which could answer this clinical question in a straightforward manner. However, this manuscript has obvious limitations and issues to be solved.

1, To answer a certain clinical question by a retrospective analysis, one needs to have a clear hypothesis BEFORE the start of the study. What was it in this study? Please describe it clearly.

2, Then, the researcher(s) must calculate the sample size depending on the hypothesis according to previous similar reports. Please describe the calculation in this study.

3, The blood loss should be estimated not only the drainage output but also calculated by the differences between preoperative and postoperative Hb, HTC, and RBC. Moreover, the estimated blood loss can be calculated by established formulae using those values. Please show these results.

6. PLOS authors have the option to publish the peer review history of their article (what does this mean?). If published, this will include your full peer review and any attached files.

Reviewer #1: **Yes: **Mohamed Abdel-Wanis

Reviewer #2: No

---

## [Author Response · Author response to Decision Letter 0]

6 Jul 2021

The authors wish to thank for a kind and meaningful review. We did our best to fulfill all the requirements and address comments properly. All the reviewer comments were taken into account. 

Journal Requirements:

 Please provide additional details regarding participant consent. 

Patients provided an informed written consent to allow their medical data to be analyzed in this study. This study was approved by the local Bioethics Committee (issued 25.11.2009, NKEBN/332/2009). – This statement is included now in the Material and Methods section. I am sorry to have not included it previously as it should have been. 

Reviewer #1: General comments

As long as wound infiltration with local anesthetics is no longer necessary for early postoperative pain control after development of local nerve blocks (lines 70-72),\\n what is the importance of the current research?

Thank you for this comment. The idea behind this study was whether LIA with adrenaline would provide additional benefit compared to local nerve blocks by limiting perioperative blood loss and justify returning to LIA with adrenaline, disregarding the local nerve blocks. This is stated clearly in the introduction section under the hypothesis. 

Abstract

. Line 23: meaning of the abbreviations Hb, HTC or RBC \\nwere not previously reported - corrected

. Lines 24-25: “amount of blood transfusions on the 1st, 4th or at patients’ discharge between infiltrated and not infiltrated groups”. \\nshould be corrected to” the 1st or 4th post-operative days” – corrected

Introduction

. Introduction must highlight the importance of the current research. The authors in lines 42-44: In total hip arthroplasty, local infiltration analgesia with adrenaline has proven to be \\nbeneficial in postoperative pain control [7,12], but it failed to show a consistent effect on bleeding [3,13,14]. \\nSo, what is the clinical significance of the current research? – other studies used different solutions and application protocols as in this study

 Material and Methods

The 2 groups are not operated by the same surgeon and this is one of the limitations of the study, it is not a randomization and not a point of strength - this expression has been removed; the discussion section explaining this matter has been updated to discuss this limitation.

Line 99: Meaning of abbreviation “BMI” was not previously mentioned. – corrected for “body mass index (BMI)”

Results

. Line 137: “There no significant differences in the female to male ratio in both groups (2x2 chi-square test “\\n

. Authors did not report use of Chi – square test in Statistical methods (lines 130-133) – the use of this test is mentioned in this section

. Line 169: “Ratio = drainage output volume (ml)/acetabular size (mm)” Was this equation developed by the authors? \\n – the formula was developed by the authors, this fact is mentioned in the text now

Discussion

Is too long and should be shortened – the discussion was updated and redundant information excluded.

. Line 226: “The study is not a randomized controlled study, but some criteria of randomization are met”. This sentence is not true and must be deleted. \\n – this section in the discussion has been updated. 

Tables

Considering each table as a separate entity, meaning of all abbreviations used in each table must be reported in a footnote after each table. – the footnotes were applied where appropriate

Reviewer #2: The authors evaluated whether local infiltration analgesia with bupivacaine and adrenaline reduced perioperative blood loss in THA. They compared a group with the analgesia and the other without it in a retrospective manner and found that there were no significant differences between the two groups. This is an interesting study which could answer this clinical question in a straightforward manner. However, this manuscript has obvious limitations and issues to be solved.

1, To answer a certain clinical question by a retrospective analysis, one needs to have a clear hypothesis BEFORE the start of the study. What was it in this study? Please describe it clearly.

The hypothesis is described at the end of the introduction. 

2, Then, the researcher(s) must calculate the sample size depending on the hypothesis according to previous similar reports. Please describe the calculation in this study.

The sample size was calculated according to previous reports and was considered adequately powered. 

3, The blood loss should be estimated not only the drainage output but also calculated by the differences between preoperative and postoperative Hb, HTC, and RBC. Moreover, the estimated blood loss can be calculated by established formulae using those values. Please show these results. 

The comparison of preoperative and postoperative values of Hb, HTC and RBC are provided in tables 2 and 3. The blood loss was estimated according to an appropriate formula by Meunier et al. (source cited in the article) and compared between groups in an additional table (Tab. 6). 

Sincerely ours

---

## [Decision Letter · Decision Letter 1]

20 Jul 2021

PONE-D-21-08207R1

Local infiltration analgesia with bupivacaine and adrenaline does not reduce perioperative blood loss in total hip arthroplasty.

PLOS ONE

Dear Dr. Ceynowa,

Thank you for submitting your manuscript to PLOS ONE. After careful consideration, we feel that it has merit but does not fully meet PLOS ONE’s publication criteria as it currently stands. Therefore, we invite you to submit a revised version of the manuscript that addresses the points raised during the review process.

Please revise accordingly.

We look forward to receiving your revised manuscript.

Kind regards,

Academic Editor

PLOS ONE

Journal Requirements:

Reviewers' comments:

Reviewer's Responses to Questions

**Comments to the Author**

1. If the authors have adequately addressed your comments raised in a previous round of review and you feel that this manuscript is now acceptable for publication, you may indicate that here to bypass the “Comments to the Author” section, enter your conflict of interest statement in the “Confidential to Editor” section, and submit your "Accept" recommendation.

Reviewer #3: All comments have been addressed

Reviewer #4: All comments have been addressed

2. Is the manuscript technically sound, and do the data support the conclusions?

Reviewer #3: Yes

Reviewer #4: Partly

3. Has the statistical analysis been performed appropriately and rigorously? 

Reviewer #3: Yes

Reviewer #4: I Don't Know

4. Have the authors made all data underlying the findings in their manuscript fully available?

Reviewer #3: Yes

Reviewer #4: Yes

5. Is the manuscript presented in an intelligible fashion and written in standard English?

Reviewer #3: Yes

Reviewer #4: Yes

6. Review Comments to the Author

Reviewer #3: • I see the senior surgeon of both groups is different. Are the bleeding control practices of both surgeons similar during the patients' operations? For example; In the hospital where I work for total hip replacement surgery, there are two kind of surgeons one does this surgery in half an hour and the other one in 2 hours. The approaches of these two surgeons to bleeding and acetabular reaming are completely different. For these and similar reasons, it would not be correct to compare the patients of these two surgeons in the same study. In my opinion, the fact that two different surgeons have two different applications does not mean that it is appropriate to compare these two different groups. As a result, this situation causes bias in the study. This is huge one of the limitations of the present study.

• Another limitation of the study is the significant difference in acetabular sizes between the two groups.

• Line 215: " ... randomized controlled study by Villatte et al. ... " The study about what?

Reviewer #4: Dear Authors

I congratulate you for your valuable study examining the effect of bupivacaine and adrenaline applied during total hip arthroplasty surgery on postoperative blood loss. I think some methodological adjustments are needed for your study to be published. My recommendations are as follows:

Keywords are not suitable for MESH. I recommend using “https://meshb.nlm.nih.gov/search” for keyword selection to increase the scannability of the article.

Comorbidity was not included in the exclusion criteria. An explanatory sentence might be appropriate in this regard.

The age range was kept very wide in the selection of the patients included in the study. Such a wide age range may make a difference in terms of susceptibility to coagulopathy. Analysis by age groups can improve the quality of the study.

Is there a difference between the two groups in terms of surgical time? No information was given about this. Extended surgical time is a factor that increases bleeding.

Is the content of the injected solution (The local anesthetic mixture was prepared with 100 ml saline solution with 50 mg of bupivacaine and 1 mg adrenaline) a standard protocol defined in the literature or is it the surgeon's preference?

“In the infiltrated group 8 patients and in the not infiltrated group 5 patients had their stay prolonged above Day 4.” The sentence should be in the results section.

BMI and blood loss can be compared. The correction calculation made for the acetabular size can also be made for the BMI.

7. PLOS authors have the option to publish the peer review history of their article (what does this mean?). If published, this will include your full peer review and any attached files.

Reviewer #3: No

Reviewer #4: **Yes: **Ismail Eralp Kacmaz

---

## [Author Response · Author response to Decision Letter 1]

10 Aug 2021

The authors wish to thank for a kind and meaningful review. We did our best to fulfill all the requirements and address comments properly. All the reviewer comments were taken into account. 

Reviewer #3: • I see the senior surgeon of both groups is different. Are the bleeding control practices of both surgeons similar during the patients' operations? For example; In the hospital where I work for total hip replacement surgery, there are two kind of surgeons one does this surgery in half an hour and the other one in 2 hours. The approaches of these two surgeons to bleeding and acetabular reaming are completely different. For these and similar reasons, it would not be correct to compare the patients of these two surgeons in the same study. In my opinion, the fact that two different surgeons have two different applications does not mean that it is appropriate to compare these two different groups. As a result, this situation causes bias in the study. This is huge one of the limitations of the present study.

We agree that the fact that both groups were not operated by a single surgeon is a limitation of this study, but this limitation is difficult to overcome in a retrospective analysis such as this study. The senior surgeons, as well as the whole operating team, has similar bleeding control practices, as they learned their craft and performed surgeries in the same center all their professional careers. The surgeries were performed by senior residents, who learned their basic skills, such as surgical approaches and bleeding control, from both senior supervising surgeons (KK, MT). The senior residents operated patients in both groups, therefore the operating surgeon bias is partially avoided. None of the patients in this study were operated on the supervising surgeons personally; they were the first surgical assistants and their role was to guide the residents with their experience during surgeries. The senior residents (all of them in their 4th or 5th year of residency with several hundred different surgeries performed personally) already have their bleeding practices well established, therefore the supervising surgeons’ influence on their basic performance, including bleeding control practices, is very limited. Therefore we believe that the influence of the surgeons’ experience and bleeding control practices have very little if any influence on the results of the study.

Additionally, we compared surgical time in both groups and there were no significant differences between them. This may be considered an indirect indication of similar bleeding control practices, as significant differences in this area would result in longer operating times when a more meticulous and time consuming bleeding control was done in one of the groups.

• Another limitation of the study is the significant difference in acetabular sizes between the two groups.

This is probably the most difficult limitation to overcome in this study. We did our best to at least partially overcome this problem by correcting the total drainage output by acetabular size as seen in tab. 5. Moreover, in the revised manuscript we performed a similar correction of total estimated blood loss (the most important outcome criterion in our opinion) by acetabular size to further check its influence on the results. No significant differences were found in this regard, as seen at the end of the Results section (below Tab. 6). A short comment was added in the discussion section. 

• Line 215: " ... randomized controlled study by Villatte et al. ... " The study about what? – an explanation is provided in the appropriate location

Reviewer #4: Dear Authors

I congratulate you for your valuable study examining the effect of bupivacaine and adrenaline applied during total hip arthroplasty surgery on postoperative blood loss. I think some methodological adjustments are needed for your study to be published. My recommendations are as follows:

1. Keywords are not suitable for MESH. I recommend using “https://meshb.nlm.nih.gov/search” for keyword selection to increase the scannability of the article.

The keywords were updated and checked in MESH. The terms: total hip arthroplasty, bleeding, hemostasis, local anesthesia, can be found in MESH. 

2. Comorbidity was not included in the exclusion criteria. An explanatory sentence might be appropriate in this regard.

A short explanation of patients’ preparation for surgery with regard to their general health status was added to the Materials and Methods section. Patients with general contraindications for surgery were not operated on and therefore not included in the study. 

3. The age range was kept very wide in the selection of the patients included in the study. Such a wide age range may make a difference in terms of susceptibility to coagulopathy. Analysis by age groups can improve the quality of the study.

An analysis in two age groups, that is between 60 and 69 years of age and 70-79 years of age, was performed separately. The number of patients above and below this age threshold was too limited for evaluation. A short remark about those results was added in the results as well as in the discussion section. Generally, the results were similar as in the general evaluation. To avoid too much specific data, the results in tables were added as supplementary material. 

4. Is there a difference between the two groups in terms of surgical time? No information was given about this. Extended surgical time is a factor that increases bleeding. – The operative time was included in patient comparison, no significant differences were found between groups. 

5. Is the content of the injected solution (The local anesthetic mixture was prepared with 100 ml saline solution with 50 mg of bupivacaine and 1 mg adrenaline) a standard protocol defined in the literature or is it the surgeon's preference?

This solution was prepared according to a standard developed in our institution, and is similar to the solutions used in other studies – this sentence is now included in the Material and Methods section. 

6. “In the infiltrated group 8 patients and in the not infiltrated group 5 patients had their stay prolonged above Day 4.” The sentence should be in the results section. – after careful consideration we removed this sentence completely, as this information is irrelevant to the results of the study and is not discussed at any point in the manuscript. 

7. BMI and blood loss can be compared. The correction calculation made for the acetabular size can also be made for the BMI. 

In this study, the acetabular size was the only significant difference between groups in the patients’ characteristics. There were no differences in age, weight, BMI and stem size, all of which can theoretically influence postoperative bleeding. 

The the idea behind the correction for acetabular size was to minimize the influence of the size differences between groups on the results. There were no significant differences between BMI results between groups, therefore we believe that a direct comparison of the results was enough and no correction for the BMI results was needed. 

We performed the correction as requested, but taking into account all the above mentioned arguments, decided not to include it in the main body of the manuscript. It does not change the results of the study and the conclusion in any way, and the inclusion of this information would make the already complicated manuscript even more difficult to read. The calculation is done in the Suppl. 1 data file, including statistical comparison. 

However, since the acetabular sizes may be of great importance in the results, we corrected the total estimated blood loss for acetabular size and included the results in the manuscript. This is the most important parameter in this study and we believe that this additional information may increase the value of the results.

---

## [Decision Letter · Decision Letter 2]

26 Aug 2021

Local infiltration analgesia with bupivacaine and adrenaline does not reduce perioperative blood loss in total hip arthroplasty.

PONE-D-21-08207R2

Dear Dr. Ceynowa,

We’re pleased to inform you that your manuscript has been judged scientifically suitable for publication and will be formally accepted for publication once it meets all outstanding technical requirements.

Kind regards,

Academic Editor

PLOS ONE

Additional Editor Comments (optional):

Reviewers' comments:

Reviewer's Responses to Questions

**Comments to the Author**

1. If the authors have adequately addressed your comments raised in a previous round of review and you feel that this manuscript is now acceptable for publication, you may indicate that here to bypass the “Comments to the Author” section, enter your conflict of interest statement in the “Confidential to Editor” section, and submit your "Accept" recommendation.

Reviewer #4: All comments have been addressed

Reviewer #5: All comments have been addressed

2. Is the manuscript technically sound, and do the data support the conclusions?

Reviewer #4: Yes

Reviewer #5: Yes

3. Has the statistical analysis been performed appropriately and rigorously? 

Reviewer #4: I Don't Know

Reviewer #5: Yes

4. Have the authors made all data underlying the findings in their manuscript fully available?

Reviewer #4: Yes

Reviewer #5: Yes

5. Is the manuscript presented in an intelligible fashion and written in standard English?

Reviewer #4: Yes

Reviewer #5: Yes

6. Review Comments to the Author

Reviewer #4: Thank you for the revisions. A study examining one of the important problems in hip arthroplasty. I think it will contribute to the literature.

Reviewer #5: Though article has been concluded against hypothesis, it would be benificial for the readers if they highlight more on the alternate methods to reduce blood loss at pre-op stage.

7. PLOS authors have the option to publish the peer review history of their article (what does this mean?). If published, this will include your full peer review and any attached files.

Reviewer #4: **Yes: **Ismail Eralp Kacmaz

Reviewer #5: No

---

## [Editor Report · Acceptance letter]

31 Aug 2021

PONE-D-21-08207R2 

Local infiltration analgesia with bupivacaine and adrenaline does not reduce perioperative blood loss in total hip arthroplasty. 

Dear Dr. Ceynowa:

I'm pleased to inform you that your manuscript has been deemed suitable for publication in PLOS ONE. Congratulations! Your manuscript is now with our production department. 

Kind regards, 

on behalf of

Dr. Robert Jeenchen Chen 

Academic Editor

PLOS ONE